# Lysophospholipids as Predictive Markers of ST-Elevation Myocardial Infarction (STEMI) and Non-ST-Elevation Myocardial Infarction (NSTEMI)

**DOI:** 10.3390/metabo11010025

**Published:** 2020-12-31

**Authors:** Elin Chorell, Tommy Olsson, Jan-Håkan Jansson, Patrik Wennberg

**Affiliations:** 1Department of Public Health and Clinical Medicine, Umeå University, SE-901 87 Umeå, Sweden; tommy.g.olsson@umu.se; 2Research Unit Skellefteå, Department of Public Health and Clinical Medicine, Umeå University, 90187 Umeå, Sweden; janhakan.jansson@regionvasterbotten.se; 3Department of Public Health and Clinical Medicine, Family Medicine, Medicine, Umeå University, 90187 Umeå, Sweden; patrik.wennberg@umu.se

**Keywords:** myocardial infarction, ST-elevation, non-ST-elevation, metabolomics, plasma protein, lysophospholipids, prediction, risk factors

## Abstract

The present study explored patterns of circulating metabolites and proteins that can predict future risk for ST-elevation myocardial infarction (STEMI) and non-ST-elevation myocardial infarction (NSTEMI). We conducted a prospective nested case-control study in northern Sweden in individuals who developed STEMI (N = 50) and NSTEMI (N = 50) within 5 years and individually matched controls (N = 100). Fasted plasma samples were subjected to multiplatform mass spectrometry-based metabolomics and multiplex protein analyses. Multivariate analyses were used to elucidate infarction-specific metabolite and protein risk profiles associated with future incident STEMI and NSTEMI. We found that altered lysophosphatidylcholine (LPC) to lysophosphatidylethanolamine (LPE) ratio predicted STEMI and NSTEMI events in different ways. In STEMI, lysophospholipids (mainly LPEs) were lower, whereas in NSTEMI, lysophospholipids (mainly LPEs) were higher. We found a similar response for all detected lysophospholipids but significant alterations only for those containing linoleic acid (C18:2, *p* < 0.05). Patients with STEMI had higher secretoglobin family 3A member 2 and tartrate-resistant acid phosphate type 5 and lower platelet-derived growth factor subunit A, which are proteins associated with atherosclerosis severity and plaque development mediated via altered phospholipid metabolism. In contrast, patients with NSTEMI had higher levels of proteins associated with inflammation and macrophage activation, including interleukin 6, C-reactive protein, chemerin, and cathepsin X and D. The STEMI risk marker profile includes factors closely related to the development of unstable plaque, including a higher LPC:LPE ratio, whereas NSTEMI is characterized by a lower LPC:LPE ratio and increased inflammation.

## 1. Introduction

Acute coronary syndrome is characterized by a sudden reduction in blood flow to the heart and is the leading cause of morbidity and mortality globally [1,2]. Patients with acute coronary syndrome can be diagnosed and classified into two types of myocardial infarction based on their electrocardiographic profile: ST-elevation myocardial infarction (STEMI) or non-ST-elevation myocardial infarction (NSTEMI). Typically, STEMI patients suffer from a ruptured plaque and total coronary occlusion with ensuing ischemia that affects the entire thickness of the myocardium, whereas NSTEMI patients have more stable vessel lesions associated with incomplete blood flow in the coronary artery, resulting in ischemia mainly within the inner region of the myocardium [3,4]. Gene polymorphisms linked to instability in atherosclerotic plaques have been found in STEMI patients, whereas NSTEMI patients are older, more often previously diagnosed with hypertension, diabetes, and atherosclerotic disease and are at higher risk of recurrent ischemic events [5]. However, available prognostic risk markers, such as the Framingham Risk Score [6], give a rough estimate of individual risk prediction. Therefore, it is of major interest to identify new prognostic markers for myocardial infarction, and specifically for STEMI and NSTEMI, to further optimize and personalize prevention and treatment [7].

Bioactive lipids are potent mediators of genetic and lifestyle factors that increase the susceptibility to cardiovascular disease (CVD) and complications, such as myocardial infarction [8]. Thus, lysophospholipids are of specific interest because they can exert multiple activities in blood cells and cells in the vessel wall [9] and may directly or indirectly mediate the progression of CVD [10] and also contribute to atherosclerotic plaque instability [11]. Importantly, total plasma phospholipid levels may not be sufficient to predict cardiovascular events, as the thrombogenicity and atherogenicity of phospholipids depends on their composition (i.e., head group, ester or ether backbone, and the saturation and carbon length of the attached fatty acid). Thus, the interpretation of total lipid content, even of a specific lipid subtype, can be misleading and yield contradictory results. In this study, we explored plasma metabolites, including lysophospholipids, and proteins to elucidate a myocardial risk signature that differs between STEMI and NSTEMI 5 years on average prior to an event.

## 2. Results

### 2.1. Cohort Characteristics

The characteristics of the study population are shown in Table 1. At sampling, the NSTEMI group was older than the STEMI group (47 years vs. 51 years, respectively), and the proportion of males was higher in the STEMI group (96% vs. 86%, *p* < 0.05). In addition, NSTEMI patients had significantly higher BMI (*p* = 0.004) than their control group. The mean time between the health examination and event was 4 years and 10 months (range 1.4–122.1 months) and was similar in the STEMI and NSTEMI groups. No change in lipoprotein(a) were observed between infarction groups and their matched control group (data not shown) [12].

### 2.2. Metabolomics and Protein Panel Analysis

The combined approach with GC-TOF/MS and LC-TOF/MS provided comprehensive coverage of serum metabolites with different chemical properties. From the mass spectrometry analyses, we detected 1140 putative metabolites, 169 of which were annotated, along with 92 proteins from the multiplex panel. All annotated metabolites and proteins are listed in Appendix A. No outliers were found in PCA of the complete dataset (data not shown). Because of the over-representation of men in this cohort, we performed the same analyses excluding women and found no change in risk metabolite and protein profile (data not shown).

### 2.3. Infarction Risk Metabolite Profiles

We found that the infarction-specific plasma metabolite risk profiles differed significantly between the infarction groups and their individually matched controls (OPLS-EP, CV-ANOVA, *p* < 0.002, Figure 1). Both infarction groups exhibited significantly altered lysophospholipids, branched-chain amino acids, and acylcarnitines. The lysophospholipid metabolite pattern differed between infarction groups; patients with STEMI had higher LPC:LPE ratios, whereas NSTEMIs had lower ratios than the matched controls (Figure 1). The LPC:LPE ratio that included linoleic acid (i.e., LPC:LPE(18:2)) was significantly altered in both myocardial infarction groups compared to matched controls, but in an opposite manner (Figure 2). In addition to an altered 18:2 ratio, NSTEMI had a significantly lower 16:0 ratio (i.e., LPC:LPE(16:0)). The underlying reason for the altered LPC:LPE ratio differed between infarction groups. Compared to controls, STEMI had lower levels of LPEs and LPCs, but the former to a greater extent, which causes a higher LPC:LPE ratio (Figure 2c, CV-ANOVA *p* < 0.05). In contrast, patients with NSTEMI had higher LPE levels, resulting in lower LPC:LPE ratios compared to matched controls (Figure 2d, OPLS CV-ANOVA *p* < 0.05, and *t* test *p* < 0.05). We also detected a significant difference in several platelet activating factors (PAFs) in both infarction groups compared to their matched controls. PAFs containing eicosatrienoic acid (C20:3) and palmitoleic acid (C16:1) were higher in NSTEMI, whereas PAFs containing linolenic acid (C18:3) were lower in STEMI compared to matched controls (Figure 1).

The branched-chain amino acids and acylcarnitine risk pattern also differed between the two infarction groups. In the STEMI group, we found significantly higher isoleucine/leucine levels (not separated due to confounded peaks in the mass spectra), along with their catabolic intermediate isovalerylcarnitine (C5iso-carnitine), and higher levels of acylcarnitine (C2-carnitine) and hexaonylcarnitine (C6-carnitine) compared to controls (Figure 1A). In the NSTEMI group, we found higher levels of glutamic acid, carnitine, acylcarnitines (C16:0-carnitine, C4-carnitine), and the branched-chain amino acid valine compared to controls (Figure 1B).

### 2.4. Infarction Risk Protein Profiles

Among the circulating proteins covered by the protein panel, we found a significantly altered protein risk profile for both infarction groups compared to their matched control groups (CV-ANOVA *p* < 0.05). For both infarction groups, we found significantly higher levels of interleukin-2 receptor subunit alpha (IL2-RA) and growth/differentiation factor-15 (GDF-15). In patients with STEMI, we found significantly altered proteins associated with phospholipid metabolism, specifically higher tartrate-resistant acid phosphatase type 5 (TR-AP) and secretoglobin family 3A member 2 (SCGB3A2) and lower platelet-derived growth factor subunit A (PDGF-A), compared to controls (Figure 3A). In contrast, in patients with NSTEMI, we found higher levels of proteins associated with inflammation and macrophage activation, specifically interleukin 6 (IL6), C-reactive protein (CRP), chemerin, cathepsin X and D, cluster of differentiation 163 (CD163), beta-cell activating factor (BAFF), and the chemokine c-c motif chemokine 16 (CXCL16).

In addition, the NSTEMI group had higher levels of the low-density lipoprotein (LDL) receptor, tissue factor pathway inhibitor (TFPI), tissue plasminogen activator (tPA), and von Willebrand factor (vWF) compared to controls (Figure 3B).

## 3. Discussion

This study highlights a unique circulating metabolite and protein risk signature for STEMI that includes metabolites and proteins associated with altered lysophospholipid metabolism. These metabolites and proteins have been shown to be associated with increased thrombosis and plaque development. Specifically, we show that the lysophospholipid ratio containing 18:2 fatty acid is altered in a different manner between STEMI and NSTEMI than matched controls. The differences in the risk marker profile between STEMI and NSTEMI may contribute to the development of novel risk scores for different myocardial infarction subtypes and increase our understanding of the underlying mechanisms, which can improve treatment efficacy.

It is becoming increasingly clear that dysfunctional lipid metabolism extends far beyond cholesterol and triglycerides. Lysophospholipids are potent mediators of inflammatory reactions and play an important role in the pathophysiology of acute coronary syndromes [13]. Specific phospholipid subtypes, such as LPCs, play a key role in plaque inflammation and vulnerability [11]. However, data on specific lysophospholipid subtypes are lacking in humans.

We show that prior to the myocardial infarction event, patients with STEMI had higher LPC:LPE ratios, especially those containing linoleic acid (C18:2), compared to matched controls. Previous studies have shown that the C18:2 lysophospholipids are strongly associated with blood clotting activities by increased conversion of C18:2 LPC to C18:2 lysophosphatic acids via autotaxin mediation [14]. Furthermore, earlier studies suggested an association between plaque severity and lysophosphatic acids [9]. However, these studies were performed in vitro or after plaque rupture, not prior to infarction. As lysophosphatic acids are imbedded in plaques/foam cells or produced during vascular injury post-infarction/plaque rupture, its precursors, such as LPC and LPE, should be more relevant predictors of risk. This finding is in line with recent data indicating an association between circulating LPCs and atherosclerotic plaque instability via platelet activation and aggregation in both mice and humans [11]. We show that both LPCs and LPEs are lower in STEMI but LPCs to a greater extent than LPEs. This is what causes the higher LPC:LPE ratio in patients with STEMI compared to matched healthy controls. Additionally included in the STEMI-risk profile was lower PAF(18:3) compared to controls, which may relate to increased vessel uptake and aggregation.

The lysophospholipid findings are supported in the protein risk profile, in which patients with STEMI exhibit alteration of phospholipid-mediated proteins associated with increased calcification and the number of diseased vessels (TR-AP) [15], inhibition of lysophospholipid metabolism (SCGB3A2) [16], and plaque formation (PDGF-A) [17]. Thus, our observation of lower LPCs and LPEs in STEMI compared to controls, specifically those containing C18:2 fatty acid, could be related to an increased conversion of LPCs and LPEs to lysophosphatic acids via autotaxin and be a sign of plaque formation and plaque severity/instability. Notably, we did not measure circulating lysophosphatic acids, which should be explored in future studies.

Lysophospholipids can be generated from glycerolipid hydrolysis, via cholesteryl esterases, and this process is greatly enhanced by the presence of bile acids [18]. In agreement with this, we show that STEMI has higher levels of secondary conjugated bile acid taurodeoxycholic acid (TDCA). In addition, endogenous bile acid has been shown to modulate cell death by interrupting apoptotic pathways, and bile acid supplementation has been shown to reduce infarct size and improve neurological function in rodent brain models [19]. Thus, the higher TDCA and betaine could be an attempt to protect against bile-induced apoptosis via inhibition of the proapoptotic mitochondrial pathway [20].

In contrast to STEMI, patients with NSTEMI had higher levels of LPCs and LPEs in addition to higher levels of proteins associated with inflammation and macrophage activation (i.e., IL6, CRP, chemerin, cathepsin X and D, CD163, BAFF, and CXCL16). Together with the higher BMI in patients with NSTEMI, this finding indicates obesity-driven mechanisms that include increased circulating low-grade inflammation and macrophage activation in adipose tissue [21].

Phospholipid homeostasis, specifically the ratio between ethanolamine- and choline-containing phospholipids, has implications in vascular wall function, as well as tissue lipid mobilization and calcium homeostasis [22]. Interestingly, both LPC and LPE have been shown to be altered in patients with angina or myocardial infarction, especially those with an increased inflammatory profile [23]. As LPC and LPE are synthesized in parallel and share enzymatic steps, an altered ratio indicates altered phospholipid metabolism and is highly associated with various metabolic disorders, including atherosclerosis [24]. Altered lysophospholipids may reflect effects on energy metabolism, steatosis, and membrane integrity, as their precursors are vital components in cell membranes and required for very low-density lipoprotein (VLDL) secretion [24]. Obesity and an oversupply of fatty acids are also suggested to contribute to a skewed phospholipid ratio, which could explain, at least in part, the higher levels of lysophospholipids observed in the more obese patients with NSTEMI compared to controls.

Concurrent with higher levels of lysophospholipids, we show higher sphingosine-1-phosphate (S1P) in NSTEMI, which regulates the important functions in cardiac and vascular homeostasis. The majority of plasma S1P seems to originate from high-density lipoprotein (HDL), and the S1P content in HDL is associated with various cardio-protective effects [25]. In contrast, S1P has been shown to increase during vascular calcification, which could be prevented by inhibition of S1P formation [26]. Interestingly, myocardial cells express both enzymes and receptors involved in sphingolipid metabolism and, thus, are able to synthesize and respond to sphingolipid metabolites, such as S1P. However, sphingolipids seem to be involved in many different pathophysiological processes that directly or indirectly affect cardiovascular functions [27].

We also show that NSTEMI has higher levels of valine, a branched-chained amino acid commonly associated with obesity [28]. Glutamic acid is also significantly higher in patients with NSTEMI, which can indicate an increased conversion of glutamine to glutamic acid to maintain nicotinamide adenine dinucleotide-H (NADH) levels and TCA intermediates and compensate for ischemic/anoxic events prior to infarction [29].

We found higher acylcarnitine levels in both infarction groups, which mainly reflects alteration of transport acyl compounds. Trans-membrane transportation of most fatty acids, and some other metabolites, occurs mostly via the carnitine shuttle, which regulates the flux of CoA-activated metabolites via acylcarnitines into the mitochondria, where they undergo β-oxidation [30]. As acylcarnitines are found outside tissues, this suggests that their efflux may serve as a detoxification process to increase metabolic flexibility and prevent CoA-trapping, allowing CoA-dependent metabolic processes to continue, including the tricarboxylic acid cycle (TCA) and β-oxidation [31,32].

We acknowledge that the fasted metabolite signatures are highly confounded by numerous lifestyle factors, and all results from this study should be validated in an independent cohort. The relatively young study group (mean age 44 years at diagnosis) may restrict the generalizability of our findings but it may also imply less confounding, in comparison with an older age group, due to coexisting disorders. Our results should also be further explored in patients with chronic coronary syndrome. Notably, male individuals were over-represented (82% male) among subjects with myocardial infarction in northern Sweden [12] during this sampling period, which is also reflected in this material. Statin usage is a potential confounder when studying circulating lipids and metabolites. However, this is unlikely in our study cohort that had a very low statins usage (1.4%) [33].

Our results suggest that further lipid screening should focus on phospholipids and lysophospholipid-related lipids and lipoproteins to increase our understanding of underlying mechanisms and improve risk assessment and treatment efficacy. In addition, future analyses should also incorporate analyses of lipid isomerization since this is potentially of key importance for their ability to interact with receptors and biological activity. This study also highlights the vast complexity and multiple mechanisms underlying myocardial infarction and the need to consider a risk profile rather than single markers to reduce false-negatives, improve risk assessment, and reduce mortality.

## 4. Materials and Methods

The study population was derived from the Västerbotten Intervention Programme (VIP) [34] and the WHO’s Multinational Monitoring of Trends and Determinants in Cardiovascular Disease (MONICA) study in northern Sweden [35]. Both VIP and MONICA included health examination programmes for CVD and diabetes with average participation rates of 59% and 77%, respectively. These studies have standardized the collection and storage procedures for metabolomics analyses and include approximately 100,000 individuals with blood samples stored in the Umeå Medical Biobank. The designs of VIP and MONICA were described in detail previously [34,35]. In northern Sweden (the counties of Västerbotten and Norrbotten), all myocardial infarction events among those aged 18–64 years between 1985 and 1999 were registered in the population-based MONICA registry (Figure 4). The myocardial infarction events were verified according to standardized WHO and MONICA criteria based on reports from hospitals or general practitioners, as well as hospital discharge records and death certificates [35]. Only prospective cases of first myocardial infarction occurring after the health examination were included in the present study. Data collection included information on medical history, symptoms, examinations, and presenting ECG. ECGs were evaluated according to Minnesota code [36], and the type of myocardial infarction was classified as STEMI or NSTEMI. If no ECG was available or the ST segment could not be evaluated (e.g., in the presence of a bundle branch block or a pacemaker), the case was classified as uncodable and not included in this study.

We identified 469 participants (Dnr: 05-14M) with available plasma samples who developed a first-ever myocardial infarction after participation in population-based health examinations. According to ECG, 52% of the men and 48% of the women had STEMI, and 29% of the men and 28% of the women had NSTEMI. The remaining cases were uncodable. From this population, we selected the 50 youngest participants who developed STEMI and the 50 youngest who developed NSTEMI. For each case, one control was randomly selected from the participants in the same health examination as the case (VIP or MONICA) and individually matched for sex, age (±2 years), date of health examination (±4 months), and geographic region. As with the cases, the exclusion criteria for the control group were previous myocardial infarction or stroke and cancer. Controls were also excluded if they had died or moved out of the region before the date of diagnosis of the myocardial infarction for the corresponding case.

### 4.1. Baseline Measurements

Smoking habits were classified into “current daily smoking” or “nonsmoking” (including previous smokers and occasional smokers). Diabetes was defined as self-reported disease from the questionnaire or fasting glucose ≥7 mmol/L and/or a 2-h post-load plasma glucose ≥11.0 mmol/L (≥12.2 mmol/L in VIP, as capillary plasma was drawn). Education level was categorized into primary education or higher. Body mass index (BMI) was calculated as weight in kg/squared height in meters (kg/m^2^).

Venous blood samples were collected in evacuated glass tubes containing heparin using standardized procedures. The samples were centrifuged at 1500 g for 15 min to obtain heparinized plasma. In the VIP and MONICA cohorts, samples were collected in the morning, after at least 4 h of fasting. Cholesterol in venous blood at the time of the health examination was measured using enzymatic methods (Reflotron bench top analysers, Boehringer Mannheim GmbH, Mannheim, Germany). All samples were initially stored at −20 °C (<7 days) and then −80 °C at the Northern Sweden Medical Research Bank until metabolomics and protein analyses. ELISAs were performed to measure CRP (IMMULITE, Diagnostic Products Corporation, Los Angeles, CA, USA). ApoA1 and B were determined by immunoturbidimetry with reagents from Dako (Glostrup, Denmark) and a calibrator (X 0947) on a Hitachi 911 multianalyzer (Roche Diagnostics GmbH, Mannheim, Germany). The apoB/apoA1 ratio was calculated. Lipoprotein (a) were measured via an enzyme immunoassay (ELITEST-Lp [a], Paris, France) and an assay standards (Hyphern BioMed, Paris, France) antibody for detection.

### 4.2. Metabolomics Analysis

In view of known methodological biases, all samples were prepared and analyzed in a specific order, keeping paired samples close, but with a randomized internal order, and analytical batches were balanced according to the blinded infarction groups [37]. Quality control samples pooled from all samples were included in each batch to assist with the identification of metabolites and to monitor instrument stability, exclude background features, and minimize methodological biases that could interfere with interpretation of the results.

The multiplatform metabolomics analyses included gas chromatography coupled with time-of-flight mass spectrometry (GC-TOF/MS) and liquid chromatography coupled with time-of-flight mass spectrometry (LC-TOF/MS, operating in positive and negative ion modes). Criteria set by the Human Metabolome Database (www.HMDB.ca) were used to assign extracted putative metabolites to different compound classes (e.g., amino acids, carbohydrates, acylcarnitines, fatty acids, and lipid subtypes). Lysophospholipid ratios were calculated from lysophosphatidylcholines (LPCs) and lysophosphatidylethanolamines (LPEs) with identical fatty acid composition [24]. Detailed sample preparation, the use of internal standards, analysis protocols, and subsequent data processing methods are provided in the Appendix A.

### 4.3. Protein Analysis

Proteins were measured using the proximal extension assay (PEA) technique [38] on a commercial proteomics array with 92 pre-selected proteins known or suggested to be involved in CVD (Olink Proseek Multiplex CVD I, Olink, Uppsala, Sweden). PEA is a homogeneous assay that utilizes pairs of antibodies equipped with DNA reporter molecules. When bound to their correct targets, the pairs give rise to new DNA amplicons that are subsequently quantified using a Fluidigm BioMark HD real-time PCR platform. Data are normalized and transformed using internal extension controls and inter-plate controls to adjust for intra- and inter-run variation [38]. A complete list of all proteins included in this study is found in Appendix A.

### 4.4. Statistical Analysis

All multivariate analyses (principal component analysis (PCA), orthogonal partial least squares (OPLS) and its extensions) [37] were performed using the software package SIMCA version 15.02 (Umetrics, Umeå, Sweden). All comparisons were made group-wise; each myocardial infarction type was compared to individual age and sex-matched controls. To validate the multivariate models, *p*-values for the differences between the predefined classes (i.e., infarction group compared to their matched control group) were calculated by analysis of variance (ANOVA) based on the cross-validated OPLS scores (CV-ANOVA) [39]. *p* < 0.05 was considered significant. Special consideration was made to ensure proper cross-validation groups (i.e., paired samples were kept in the same CV group) and reduce the risk of creating over-fitted models. If not stated otherwise, a metabolite was considered to contribute significantly to the metabolite profile if it was significantly altered according to the multivariate confidence interval based on jack-knifing [40] and a significant univariate *p*-value on a 95% significance level. Univariate *p*-values were calculated using a paired *t* test on the putative metabolites highlighted by the multivariate models, as well as for the baseline clinical characteristics.

## 5. Conclusions

We show that STEMI and NSTEMI differ in their circulating metabolite and protein risk signatures. Individuals that will develop a STEMI have higher LPC:LPE ratios containing C18:2 fatty acid (linoleic acid), whereas the opposite is found in NSTEMI. Lysophospholipids actively mediate instable plaque formation [11], which is in line with the finding of culprit plaques in STEMI [41]. This is strengthened by our protein risk profile for STEMI, which included proteins highly associated with phospholipid metabolism, as well as increased thrombosis and plaque severity. In contrast, NSTEMI has a biochemical risk profile in line with their higher BMI compared to controls, which includes higher lysophospholipids, a branched-chain amino acid (valine), inflammatory markers, and proteins associated with increased macrophage infiltration.

## Figures and Tables

**Figure 1 metabolites-11-00025-f001:**
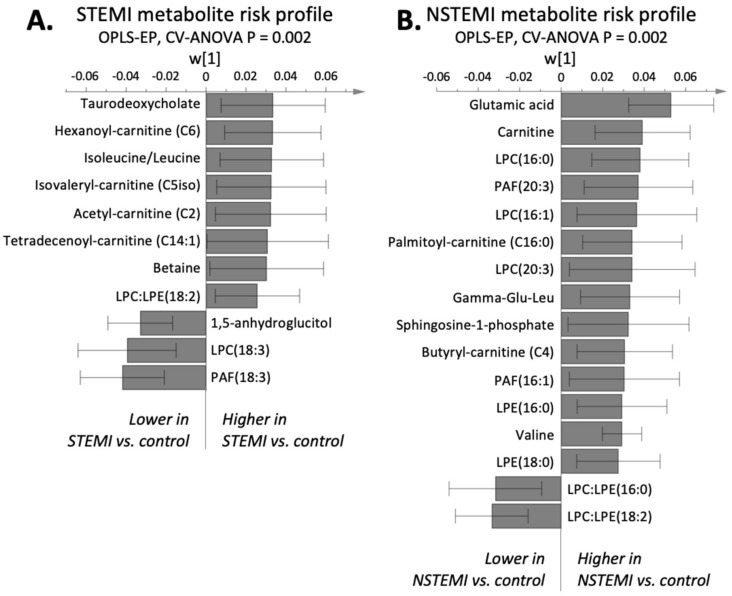
Infarction-specific multivariate models (OPLS-EP) calculated from plasma metabolomics. Annotated parameters that were significantly (*p* < 0.05) different between cases and their matched controls are included. (**A**) STEMI metabolite risk profile. (**B**) NSTEMI metabolite risk profile. The multivariate confidence intervals indicate significance on a 95% based on jack-knifing.

**Figure 2 metabolites-11-00025-f002:**
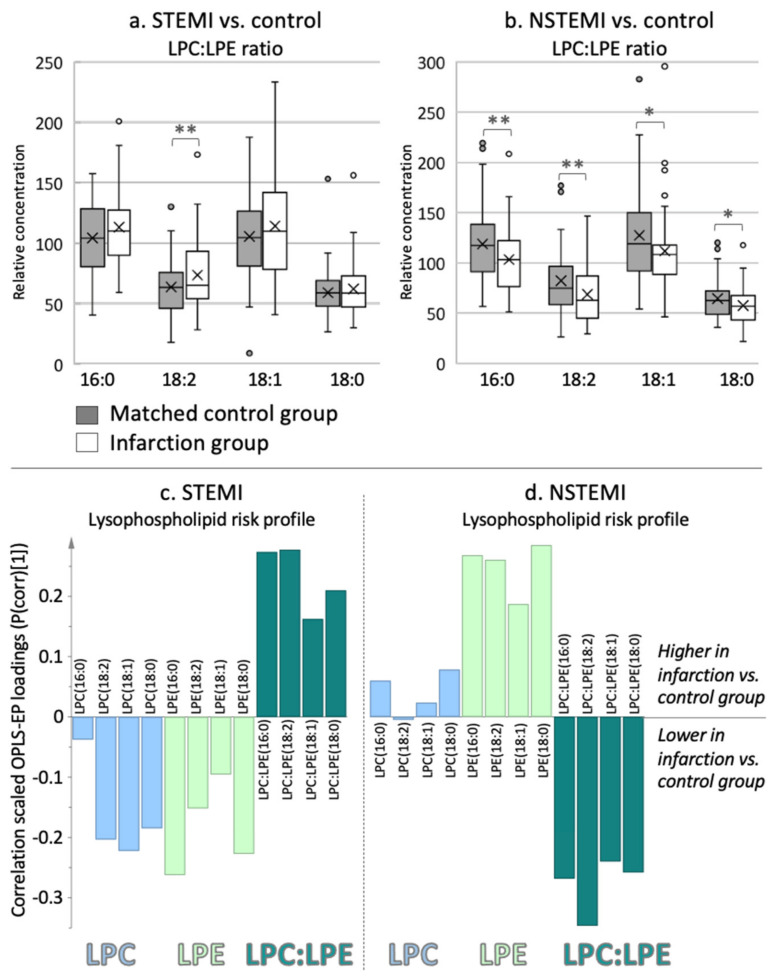
Raw data describing the relative concentration of lysophospholipid ratios with different fatty acid composition. (**a**) STEMI patients (white bars) and (**b**) NSTEMI patients (white bars) compared to their respective controls (grey bars). LPC, lysophosphatidylcholine; LPE, lysophosphatidylethanolamine. ** Ratios that are significantly altered between cases and controls in multivariate models and univariate *t* tests. * Ratios that are significantly altered only in univariate analysis based on a 95% confidence level. Confidence intervals are calculated on a 95% confidence levels using Student’s t-test. (**c**) The higher LPC:LPE ratios in STEMI are caused by lower LPCs and especially LPEs. (**d**) Lower LPC:LPE ratios in NSTEMI are caused by higher LPEs and no difference in LPCs compared to matched controls (CV-ANOVA *p* < 0.05).

**Figure 3 metabolites-11-00025-f003:**
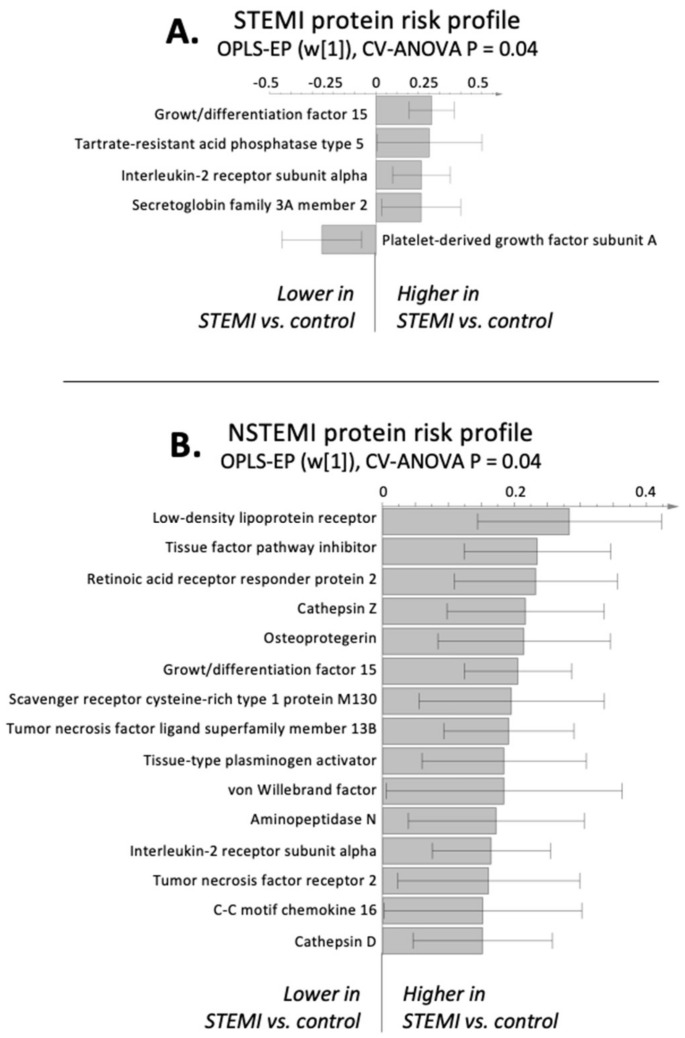
Infarction-specific multivariate models (OPLS-EP) calculated from the multiplex protein panel. Only proteins that are significantly (*p* < 0.05) altered between cases and their matched controls are shown. (**A**) STEMI protein risk profile. (**B**) NSTEMI protein risk profile. The multivariate confidence intervals indicate significance on a 95% based on jack-knifing.

**Figure 4 metabolites-11-00025-f004:**
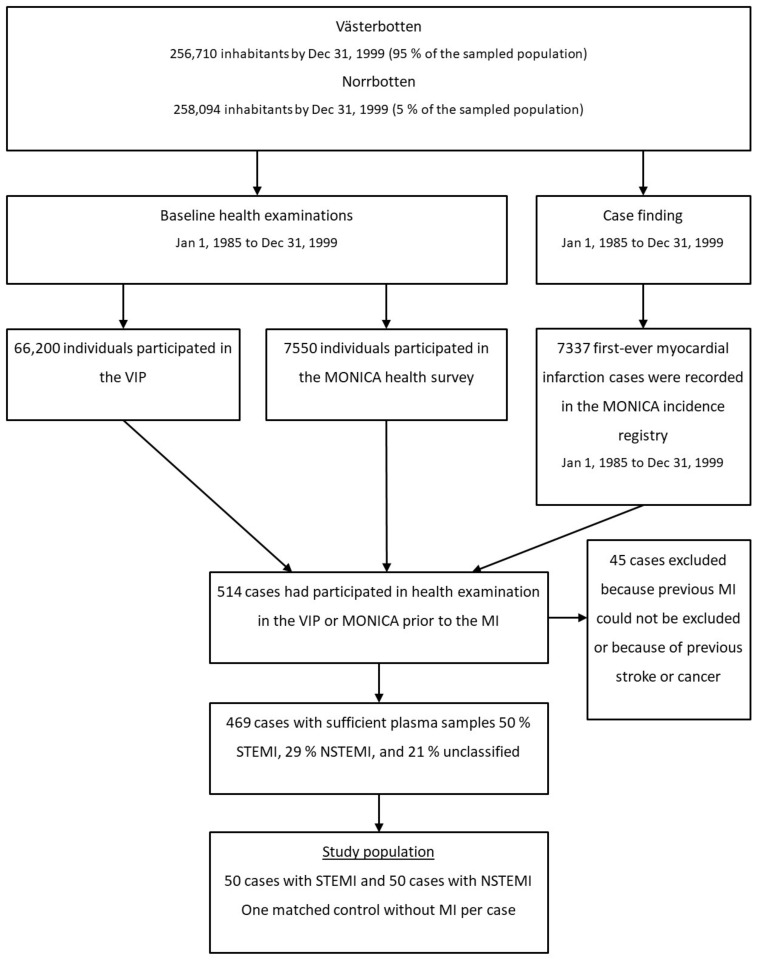
Flow chart of the study population.

**Table 1 metabolites-11-00025-t001:** Baseline characteristics of the cases and referents with ST-elevation myocardial infarction (STEMI) and non ST-elevation myocardial infarction (NSTEMI).

	STEMI			NSTEMI		
	Cases (*n* = 50)	Referents (*n* = 50)	*p*-Value	Cases (*n* = 50)	Referents (*n* = 50)	*p*-Value
Age at screening, years	42.0 (5.2)	42.1 (5.5)	0.91	45.8 (5.1)	45.8 (5.1)	0.99
Age at MI/SCD, years	46.8 (5.5)			50.6 (5.5)		
Sex (% male)	96	96	1	86	86	1
Smoking (%)	48	10.4	<0.001	42.9	15.2	0.003
Diabetes (%)	10	0	0.02	6	4	0.64
Hypertension (%)	54	36	0.07	46	26	0.04
Low level of education (%)	32	36	0.67	42.6	25.5	0.08
BMI, kg/m^2^ ^a^	26.6 (4.0)	25.5 (3.8)	0.25	27.1 (4.5)	25.3 (4.0)	0.004
SBT, mmHg	135.8 (14.8)	131.2 (14.9)	0.12	132.7 (13.3)	128.9 (12.2)	0.15
DBT, mmHg	88.4 (11.2)	83.1 (9.8)	0.01	87.3 (9.0)	82.5 (9.3)	0.013
Cholesterol, mmol/L	6.27 (1.19)	5.97 (1.24)	0.21	6.68 (0.99)	6.13 (1.19)	0.015
Glucose, mmol/L ^a^	5.10 (0.90)	5.16 (0.63)	0.87	5.10 (1.25)	5.28 (0.63)	0.69
CRP, ng/L ^a^	1.15 (1.85)	1.24 (2.34)	0.81	2.29 (3.31)	1.06 (1.11)	0.004
ApoB/apoA1 ratio	0.99 (0.33)	0.79 (0.21)	0.001	1.00 (0.24)	0.82 (0.26)	<0.001

MI, myocardial infarction; sudden cardiac death (SCD); BMI, body mass index; SBT, systolic blood pressure; SBT, systolic blood pressure; DBT, diastolic blood pressure; CRP, C-reactive protein. Data are given as proportions (%), mean value and standard deviation, or ^a^ median and interquartile range. Low level of education refers to 9 years of compulsory education.

## Data Availability

Data will be available on request.

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
