# Peer review of "Lysophospholipids as Predictive Markers of ST-Elevation Myocardial Infarction (STEMI) and Non-ST-Elevation Myocardial Infarction (NSTEMI)"

_metabolites, 2020, doi:10.3390/metabo11010025_

Round 1

Reviewer 1 Report

Chorell et al. described the lysophospholipids, especially LPC/LPE ration as markers of STEMI and NSTEMI. Among lipid data, authors indicated that 18:2LPC/LPE ratio is an important key factor. This study provides useful information for STEMI and NSTEMI studies and therapy. There are a number of issues to be addressed to strengthen the conclusion and improve the quality.

*Profiles and amount of free fatty acids and triglycerides are also important factors. Authors had better show them.

*C18:2 was changed in LPC/LPE ratio. Are C18:2 changed in the other lipids such as phospholipids, cholesterols and triglycerides?

*PAF(C2:0) is a potent lipid mediator. Dose PAF(16:1) and PAF(18:3) possess the activity? If some literatures indicated, they had better be referred.

*I could not open “Table S1” file, so I did not read molecular lists. Additionally, Supplementary material did not show the references. Is “A et al. 2005” a mistake?

Author Response

Chorell et al. described the lysophospholipids, especially LPC/LPE ration as markers of STEMI and NSTEMI. Among lipid data, authors indicated that 18:2LPC/LPE ratio is an important key factor. This study provides useful information for STEMI and NSTEMI studies and therapy. There are a number of issues to be addressed to strengthen the conclusion and improve the quality.

 *Profiles and amount of free fatty acids and triglycerides are also important factors. Authors had better show them.

REPLY: We thank the reviewer for this suggestion. Unfortunately, we do not have data on triacylglycerides since lipidomics were not included in these analyses. This has also been addressed in the discussion (page 7, lines 236-238): “Our results suggest that further lipid screening should focus on phospholipids and lysophospholipid-related lipids and lipoproteins to increase our understanding of underlying mechanisms and improve risk assessment and treatment efficacy”. Regarding fatty acids, all fatty acids that were significant altered between infraction groups and their respectively controls are shown in figure 1. A full description of all detected fatty acids is found in Table S1. Of note, the included mass spectrometry-based metabolomics analyses will include lysophospholipids since these lipids are more polar and included in the metabolite extraction. To extract larger lipids, such as triacylglycerides and phospholipids, a different extraction procedure is required. 

*C18:2 was changed in LPC/LPE ratio. Are C18:2 changed in the other lipids such as phospholipids, cholesterols and triglycerides?

REPLY: We thank the reviewer for this valid and very interesting question. However, to obtain fatty acyl composition of lipid sub-species this requires an additional lipid extraction and full lipidomics analysis as we replied in the previous comment. In future studies we will include and complement metabolomics with lipidomics analyses when possible.

*PAF(C2:0) is a potent lipid mediator. Dose PAF(16:1) and PAF(18:3) possess the activity? If some literatures indicated, they had better be referred.

REPLY: We are not sure what the reviewer is asking for since we could not identify PAF(C2:0) in the lipid literature. Please let us know if the reviewer refers to inositoylated Platelet-Activating Factor (Ino-C2-PAF)? This lipid has been highlighted as an active signaling molecule. Unfortunately, the analyses included in this study were not suitable for detecting this compound. This is indeed an interesting compound and we have incorporated in into our standard lipid library and will be included in future analyses.

*I could not open “Table S1” file, so I did not read molecular lists. Additionally, Supplementary material did not show the references. Is “A et al. 2005” a mistake?

REPLY: We are not sure why Table S1 was not readable and are sorry for this. During future communication with the editors, we will make sure that this file is fully functional. The references “A et al 2005” have been added to the supplementary material. Notable. A is the surname of the first author of this paper.

Reviewer 2 Report

This is a nice study exploring new marker of acute event in ACS population

However, the following issues should be addressed:

  1. A matched control of chronic coronary syndrome (CCS) patients is mandatory since stable ahterosclerosis maybe already associated to an LPC:LPE imbalance
  2. Correlation with Lp(a) should be provided
  3. No data are provided on the previous and ongoing therapy. Is there any effect from statin therapy and type of lipid lowering agents?
  4. Is there any change in LPC:LPE profile at 30 day follow up according to the diagnosis at admission?  

Author Response

This is a nice study exploring new marker of acute event in ACS population

However, the following issues should be addressed:

  1. A matched control of chronic coronary syndrome (CCS) patients is mandatory since stable ahterosclerosis maybe already associated to an LPC:LPE imbalance

REPLY: We agree with the reviewer that the addition of a control group with stable coronary artery disease would have been highly interesting. However, this study is based on a prospective nested case-referent design where all participants in the present study were included in a primary care setting without previously known myocardial infarction or stroke. We have no data on chronic coronary artery disease at the time of blood sampling.

  1. Correlation with Lp(a) should be provided

REPLY: We thank the reviewer for highlighting this. We found no difference between NSTEMI or STEMI individuals and their controls regarding Lp(a) levels. In accordance with the reviewer comment this was added to the method and result section (page 2, line 70-71) along with a reference to a previous published study from the same population that showed no associated between myocardial infarction and Lp(a).  

  1. No data are provided on the previous and ongoing therapy. Is there any effect from statin therapy and type of lipid lowering agents?

REPLY: We fully agree with the reviewer that the use of statins is of importance. Unfortunately, we have no data on statin use of the included individuals. However, the use of statins in the overall population in Northern Sweden from which our cohort were sampled from was very low during the sampling period (1994), around 1.4%. In line with reviewer comment, we added a comment and reference of general statin use of this population in the discussion (page 7, lines 233-235).

  1. Is there any change in LPC:LPE profile at 30 day follow up according to the diagnosis at admission?  

REPLY: Unfortunately, we cannot investigate this since we have no blood samples available after cardiac event/follow-up. 

Reviewer 3 Report

Congratulations. This is an excellent contribution. I read the paper and consider it a great contribution to the field, withing its own limitations (young average age of the cohort, mainly males).

Author Response

Congratulations. This is an excellent contribution. I read the paper and consider it a great contribution to the field, withing its own limitations (young average age of the cohort, mainly males).

REPLY: We thank the reviewer for these encouraging words!

Round 2

Reviewer 1 Report

Chorell et al. indicated that 18:2LPC/LPE ratio is an important key factor. This revised manuscript was well answered. My previous comment means indicated below.

My previous comment

*PAF(C2:0) is a potent lipid mediator. Dose PAF(16:1) and PAF(18:3) possess the activity? If some literatures indicated, they had better be referred.

PAF(C2:0) means PAF containing acetic acid (C2:0) at the sn-2 position. PAF is biosynthesized from lyso-PAF and acetyl-CoA (C2:0-CoA). This type of PAF has potent activity. Did authors mean that PAF (C16:1) and PAF(C18:3) have C16:1 and C18:3 at the sn-1 position? If so, I misunderstood and authors do not need to revise.

Author Response

We thank the reviewer for a clear reply. Unfortunately, our analyses cannot discriminate between sn1 and sn2 positions of PAFs. However, we fully agree with the reviewer that this is of importance to understand and highlight important mediators since isomerization will influence lipids ability to interact with receptors and thus biological activity. In accordance with the reviewer comment, this has been added to the discussion (page 7, lines 239-241)

Reviewer 2 Report

This article can be considered a good paper that should be improved with additional experiments as indicated im my previous revision. It can be a suggestion for a new investigator protocol

Thus, it can be accepted by improving the limitation section based on my concerns

Author Response

In accordance with the reviewer comment, we addressed the importance for further validation of our results in other cohorts and in particular in patients with chronic coronary syndrome in the discussion (page 7, line 231-232).